# Detection of a Novel *MSI2-C17orf64* Transcript in a Patient with Aggressive Adenocarcinoma of the Gastroesophageal Junction: A Case Report

**DOI:** 10.3390/genes14040918

**Published:** 2023-04-15

**Authors:** Anna Ferrari, Roberto Fiocca, Elena Bonora, Chiara Domizio, Eugenio Fonzi, Davide Angeli, Gian Domenico Raulli, Sandro Mattioli, Giovanni Martinelli, Chiara Molinari

**Affiliations:** 1Biosciences Laboratory, IRCCS Istituto Romagnolo per lo Studio dei Tumori (IRST) “Dino Amadori”, 47014 Meldola, FC, Italy; 2Unit of Anatomic Pathology, Ospedale Policlinico San Martino IRCCS, 16125 Genova, Italy; 3Department of Surgical and Diagnostic Sciences (DISC), University of Genova, 16125 Genova, Italy; 4Department of Medical and Surgical Sciences (DIMEC), Alma Mater Studiorum, University of Bologna, Via Massarenti 9, 40126 Bologna, Italy; 5Department of Life Sciences and Biotechnology, Ferrara University, 44124 Ferrara, Italy; 6Unit of Biostatistics and Clinical Trials, IRCCS Istituto Romagnolo per lo Studio dei Tumori (IRST) “Dino Amadori”, 47014 Meldola, FC, Italy; 7Pathology Unit, AUSL della Romagna, 48121 Ravenna, Italy; 8GVM Care & Research Group, Division of Thoracic Surgery-Maria Cecilia Hospital, 48022 Cotignola, RA, Italy; 9Alma Mater Studiorum-University of Bologna, 40126 Bologna, Italy; 10Scientific Directorate, IRCCS Istituto Romagnolo per lo Studio dei Tumori (IRST) “Dino Amadori”, 47014 Meldola, FC, Italy

**Keywords:** gastroesophageal cancer, chemotherapy resistance, poorly-cohesive, fusion, *MSI2*

## Abstract

Adenocarcinoma of the esophagus (EAC) and gastroesophageal junction (GEJ-AC) is associated with poor prognosis, treatment resistance and limited systemic therapeutic options. To deeply understand the genomic landscape of this cancer type, and potentially identify a therapeutic target in a neoadjuvant chemotherapy non-responder 48-year-old man, we adopted a multi-omic approach. We simultaneously evaluated gene rearrangements, mutations, copy number status, microsatellite instability and tumor mutation burden. The patient displayed pathogenic mutations of the *TP53* and *ATM* genes and variants of uncertain significance of three kinases genes (*ERBB3*, *CSNK1A1* and *RPS6KB2*), along with *FGFR2* and *KRAS* high copy number amplification. Interestingly, transcriptomic analysis revealed the Musashi-2 (*MSI2)-C17orf64* fusion that has never been reported before. Rearrangements of the RNA-binding protein *MSI2* with a number of partner genes have been described across solid and hematological tumors. *MSI2* regulates several biological processes involved in cancer initiation, development and resistance to treatment, and deserves further investigation as a potential therapeutic target. In conclusion, our extensive genomic characterization of a gastroesophageal tumor refractory to all therapeutic approaches led to the discovery of the *MSI2-C17orf64* fusion. The results underlie the importance of deep molecular analyses enabling the identification of novel patient-specific markers to be monitored during therapy or even targeted at disease evolution.

## 1. Introduction

Adenocarcinoma of the esophagus (EAC) and gastroesophageal junction (GEJ-AC) is an aggressive disease usually diagnosed at advanced stages, with limited curative options, a median life expectancy of 12 months and a 5-year survival rate of 12–20% in Western populations [1,2,3]. Its incidence has increased several-fold in Western countries in recent decades, being now the eighth most common cancer and the sixth leading cause of death in the world.

Several strong epidemiologic risk factors have been identified including reflux symptoms, obesity and smoking, and an etiologic role for inherited genetics more recently emerged, including candidate risk genes and pathways [4].

Patients with locally advanced or oligometastatic tumors are mostly candidates for multimodal therapy. Neoadjuvant chemotherapy or chemoradiotherapy, followed by radical surgery, is the standard of care for these patients, aiding tumor shrinkage, a higher rate of complete resection and eradication of circulating malignant cells [2]. The individual response is unpredictable, however 80% of patients achieve only an incomplete or absent response and their survival remains very poor [5,6]. In particular, poorly differentiated gastroesophageal adenocarcinomas, eventually characterized by the presence of signet ring cells (SRC), are generally resistant to current oncological therapies and are associated with poor prognosis [6]. Moreover, large-scale genomic studies have shown that EAC and GEJ-AC mainly resemble gastric cancer with chromosomal instability, though among cases not clearly of esophageal origin, positivity for microsatellite instability (MSI) and Epstein–Barr Virus (EBV) have been identified [7]. Their high mutational burden and mutation rates, the high frequency of copy number alterations and somatic structural rearrangements give rise to a significant heterogeneity among patients and within the same tumor, the latter potentially correlated to a poor response to standard chemotherapy treatments and to a worse outcome [6].

Therefore, there is an urgent need to find prognostic and predictive markers guiding the selection of the most appropriate treatments for patients affected by these tumors, also taking into consideration that, in the localized setting, novel systemic therapeutic regimens, such as immunotherapy, are underway.

While the analysis of HER2 in localized settings do not have a clear recommendation, the evaluation of MSI, EBV and PD-L1, are becoming more important. Accordingly, CLDN18.2 and FGFR might be interesting targets in the near future [8].

In this context, a wide genomic characterization of each case by next-generation sequencing (NGS) may be very helpful in the identification of biomarkers for the response to therapy, actionable alterations and new clinically-relevant targeted drugs.

Among molecular aberrations, the discovery and characterization of new fusion genes can both improve patient diagnosis and precision medicine, as it has been recently demonstrated for a number of novel rearrangements [9]. 

In the context of gastroesophageal cancer, some recurrent in-frame fusions have been associated with diffuse gastric cancer (DGC) and an aggressive disease phenotype [10]. In particular, the most common fusion, *CLDN18–ARHGAP26*, is more prevalent in early-onset DGC and correlates with poor survival and chemoresistance [11].

The aim of this report was to characterize fusion genes in gastroesophageal cancer, aiding in a clinically useful molecular refinement of this tumor subtype.

## 2. Case Presentation

A 48-year-old Caucasian patient affected by GEJ-AC (clinical stage cT3N0M0) received fluorouracil plus leucovorin, oxaliplatin, and docetaxel chemotherapy (FLOT4 regimen) followed by total gastrectomy with an extended lymphadenectomy (ypT4aN3M0 G3). As reported in Figure 1, the tumor presented two different components, including a glandular and a poorly-cohesive one. The poorly-cohesive component was prevalent (90%), with focal SRC areas (<25%). No tumor regression was observed following neoadjuvant chemotherapy (regression score 5 according to Mandard classification). Cancer recurred during adjuvant therapy and the patient died 7 months after surgery. 

The nucleic acids were extracted from the formalin-fixed paraffin-embedded (FFPE) surgical specimen, with macrodissection of all different components of the tumor before combining them together. Genetic analyses were performed by Next Generation Sequencing using the Trusight Oncology (TSO) 500 panel (Illumina, San Diego, CA, USA) on a NextSeq 550 sequencer (Illumina). TSO 500 allows detection of the mutational status of more than 500 genes, copy number variation (CNV) for 59 genes, as well as MSI and the tumor mutational burden (TMB) with an analytical sensitivity > 96% (for all variant types at 5% variant allele frequency (VAF)) and specificity > 99.9% (Illumina data sheet M-GL-00173 v4.0). Data were processed through the TSO500 Local App pipeline which provides quality metrics for each sample, TMB, MSI, small variant calling and CNV detection. To select clinically-relevant variants, we filtered those retained according to the software small variant default settings as follows: (a) we excluded variants with a population frequency higher than 0.01 into at least one of the public human polymorphism databases (esp5400, https://evs.gs.washington.edu/EVS/ accessed on 22 November 2011; ExAC, http://exac.broadinstitute.org/;GnomAD accessed on 23 April 2016, https://gnomad.broadinstitute.org/ v2.1.1, accessed on 18 March 2019); (b) we discarded synonymous, 3′/5′ untranslated region, intronic and intergenic variants; (c) we excluded variants with a coverage lower than 100× or a VAF lower than 5%; (d) we held back variants annotated by ClinVar as pathogenic or likely pathogenic and rejected benign/likely benign variants (https://www.ncbi.nlm.nih.gov/clinvar/ accessed on 16 March 2020); (e) we integrated Varsome premium information (v. 11.6.1; https://varsome.com/ accessed on 2 February 2023) in the characterization of variants not annotated or marked as “uncertain significance” by ClinVar. RNA sequencing analysis of 1385 cancer genes was performed by the TruSight RNA Pancancer Panel (Illumina) on MiSeq (Illumina). Illumina panel sensitivity tests were reported on their website. Transcriptome data were analyzed by an in-house pipeline to identify true positive transcripts. In detail, we used FusionCatcher, STAR-Fusion and two Basespace applications (RNA-Seq Alignment and TopHat Alignment; Illumina). Each tool used its own aligner except for FusionCatcher which combines BLAT, STAR, Bowtie and Bowtie2. The four tool outputs were unified and filtered by a multistep strategy to identify true positive transcripts. Briefly, retained fusions were detected by at least three tools and we introduced two further criteria to retain or reject fusions detected by two or one tool specified in patent request (PCT/EP2021/065692). 

The DNA analysis revealed very high *FGFR2* and *KRAS* gene amplifications (estimated copy number of 20.6 and 19.9, respectively; Appendix A), a status of microsatellite stability (MSI < 20%) and of a low TMB (<10 mutations/megabase). Moreover, as reported in Table 1, two pathogenic mutations have been detected: a hotspot variant in the *TP53* gene (*TP53*:p.R248W; VAF = 46.3%) and a mutation of the *ATM* gene (*ATM*: p.Gly2023Arg; VAF = 46.8%). Both variants were also detected at the RNA level. We identified three additional mutations, with uncertain significance, targeting three different kinases: Erb-B2 receptor tyrosine kinase 3 (*ERBB3*), casein kinase 1 α 1 (*CSNK1A1*) and ribosomal protein S6 kinase B2 (*RPS6KB2*; Table 1 and Appendix A).

By transcriptome sequencing we obtained 4.025.762 reads and 99.32% of them were aligned reads. Fusion analysis excluded the presence of the diffuse-type associated fusions *CLDN18–ARHGAP26* and *CTNND1–ARHGAP26* that are frequently found in SRC. Conversely, all fusion calling tools detected a novel transcript, namely Musashi-2 (*MSI2)–C17orf64* (*MSI2*, chr17:55674311, ex8; *C17orf64*, chr17:58508542, ex6; Figure 2A). The resulting in-frame chimera caused the loss of all MSI2 polyadenylate binding protein domains and of the C17orf64 DUF4208 domain (Figure 2A). The fusion transcript was confirmed by RT-PCR and Sanger sequencing (Figure 2B,C).

## 3. Discussion

The patient described in the present case report was affected by a mixed adenocarcinoma with two distinct histological components [12,13]: a poorly-cohesive one (90% of the tumor) including SRC, and a glandular one (10% of the tumor). The proportion of SRCs in poorly cohesive subtypes is a marker of differentiation able to predict tumor prognosis [14], thus suggesting a potential connection between low (<25%) SRC proportion in the analyzed tumor and the poor overall survival of the patient. Moreover, it has been reported that chemotherapy and chemoradiotherapy are more effective in intestinal type carcinomas compared with poorly-cohesive/mixed type carcinomas [5].

The resistance to conventional therapies is a recurring issue in EAC and GEJ-AC, since only a small percentage of patients achieve a complete pathological response, leading to a better survival [6]. The definition of predictive biomarkers is needed to tailor treatments for patients and to identify additional targets for novel therapeutic approaches. 

EAC and of GEJ-AC resemble chromosomally unstable gastric adenocarcinoma in their genetic makeup [7]. *TP53* is the most frequently mutated gene in these tumors, being mostly prevalent in GEJ-AC [15]. In particular, the *TP53*^R248W^ variant found in our case, is the most common site-specific mutation in all cancers. It induces loss of function by preventing p53 binding to its target promoters, leading to a consequent p53 dysfunction known to be directly related to poor response to chemotherapy [16]. The disruption of the DNA damage response pathway in the present tumor is also reinforced by the pathogenic variant found in the *ATM* gene, which could affect its expression, impacting on the patient survival [17]. Indeed, a potential benefit from combined chemotherapy and PARP-inhibitor treatment has been suggested [18]. 

Moreover, the high level of *FGFR2* amplification is of particular interest due to its association with poor patient prognosis [19] and therapeutic implications as a potential target of tyrosine kinase inhibitors [20,21]. *FGFR2* amplification in gastroesophageal cancers has been found with a frequency ranging from 2.5% to 7.4%, it is often associated with FGFR2 overexpression [22] and other receptor tyrosine kinases’ alterations, such as *KRAS* amplification. Both the level of *FGFR* copy number and the putative co-occurring alterations which can modify FGFR2-directed therapy should be considered for a rational combination of treatment options and patients’ selection.

Several fusion transcripts have been repeatedly reported as drivers of gastroesophageal cancer, being mainly described in young patients and in diffuse-type cancers and SRC carcinomas [10,11]. However, none of them have been found in the present tumor.

However, we detected a new rearrangement, involving the *MSI2* and *C17orf64* genes that, to our knowledge, has never been previously described.

Little is known about C17orf64, a poorly characterized protein showing a testis-specific expression pattern and heavy methylation of the promoter in high-grade serous ovarian carcinoma [23,24]. Conversely, MSI2 is a crucial regulator of cancer stem cell programs by acting on the stability and translation of target mRNAs that encode key proteins of oncogenic signaling pathways (e.g., TGFβR1/SMAD3, NUMB/Notch, PTEN/mTOR, MET, and MYC) [25,26,27,28,29].

MSI2 has been deeply studied across cancers. Its expression strongly correlated with poor clinical prognosis in hematological malignancies [30,31]. Moreover, it is overexpressed in many solid tumors, including glioblastoma, breast cancer, cervical cancer, pancreatic cancer, gastric cancer (GC), colorectal cancer (CRC) and hepatocellular c arcinoma (HCC) [28,32]. High *MSI2* levels were associated with poor tumor differentiation, and poor prognosis. Indeed, functional studies demonstrated a role of MSI2 in maintaining stemness properties, metastatic capacity, in vitro and in vivo tumorigenic ability and promoting drug resistance either in HCC or pancreatic tumors [33,34,35,36]. Interestingly, in pancreatic cancer, a high proportion of circulating tumor cells (CTCs) expressed Msi2 (Msi2+), and were more tumorigenic than Msi2− CTCs, posing a greater risk for tumor dissemination [33]. MSI2 is also a central component of oncogenic pathways promoting intestinal transformation [37], its expression is further elevated during CRC progression, and is associated with poor prognosis [38]. In GC the expression level of MSI2 was positively associated with invasion depth, stage, degree of differentiation and tumor size [39].

The association between MSI2 overexpression and poor prognosis has been recently established across different malignancies, thus indicating that MSI2 could be a novel prognostic biomarker and therapeutic target [40].

Finally, recent findings pointed to MSI2 as a promising therapeutic target for solid and hematological malignancies. Small molecules inhibiting its oncogenic activity are currently under preclinical investigation [41,42,43] and their potential role in treating cases carrying *MSI2* rearrangements deserves a case-by-case evaluation.

A rare *MSI2–HOXA9* translocation has been previously identified in patients progressing from chronic myeloid leukemia to blast crisis [44]. More recently, a *MSI2–PC* rearrangement has been described in myelodysplastic syndrome [45]. These fusions suggest a potential role of MSI2 as a driver of enforced expression of the partner gene.

The novel *MSI2–C17orf64* that we describe resulted in the loss of the *MSI2* C-terminal region and poly-A binding protein (PABP)-interaction domain, which regulates the translation of a subset of MSI2 target genes [28]. In this case, it is not clear whether the rearrangement activates the aberrant expression of an unexplored *C17orf64* domain or whether it generates a functional, dysfunctional or even inactive chimeric protein. Although one limitation of our work is the analysis of a single case carrying the fusion, the consequences of the rearrangement on MSI2 activity deserve further investigation.

## 4. Conclusions

This is the first case reporting an *MSI2–C17orf64* fusion in tumors in general, and in gastroesophageal adenocarcinoma, in particular. The *MSI2* gene is involved in intestinal and hematological stem cell pathways and promotes tumor progression, dissemination and drug resistance in several solid and hematological malignancies. Future studies will highlight the biological and clinical significance of this novel fusion. This finding opens an interesting new perspective on the importance of a deep genomic characterization of gastroesophageal tumors that are actually refractory to every therapeutic approach.

## 5. Patents

PCT application No. PCT/EP2021/065692 (10 June 2021): Method to identify linked genetic fusions.

## Figures and Tables

**Figure 1 genes-14-00918-f001:**
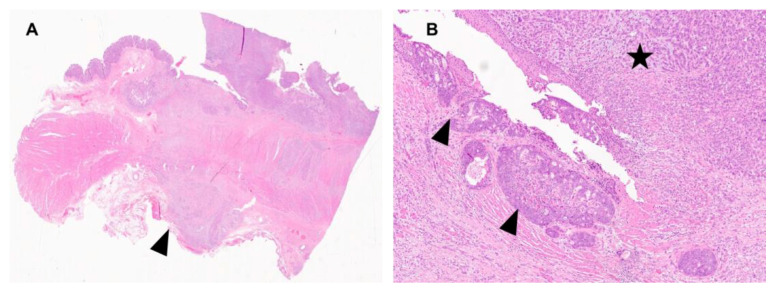
Hematoxylin and Eosin staining of the surgical specimen. (**A**) Low power view (2× magnification) showing the serosa invasion (arrowhead). (**B**) The glandular (intestinal-type) and the poorly-cohesive components are marked by arrowheads and an asterisk, respectively (10× magnification).

**Figure 2 genes-14-00918-f002:**
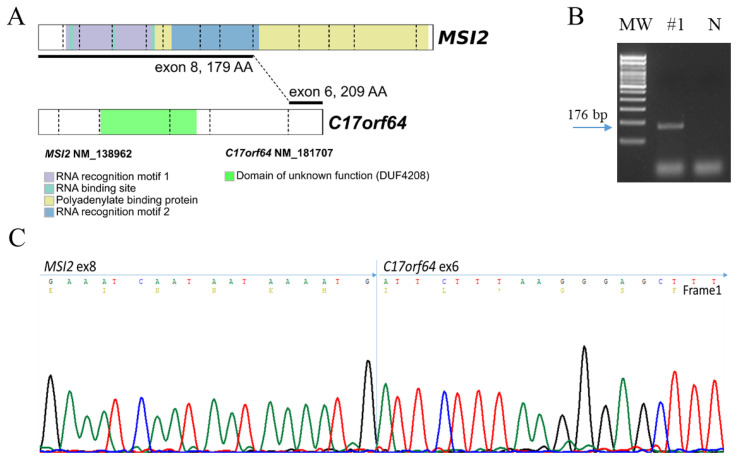
*MSI2–C17orf64* fusion. (**A**) MSI2 and C17orf64 protein diagrams, domain annotations and fusion scheme between *MSI2* exon 8 and *C17orf64* exon 6 (*MSI2*, NM_138962, chr17:55674311, +; *C17orf64*, NM_181707, chr17:58508542, +; Human hg19). (**B**) Agarose gel electrophoresis of tumor sample (#1). Lane MW: DNA marker (100 bp ladder). Samples 1: *MSI2-C17orf64* PCR amplified product; N = negative. (**C**) Sanger sequencing chromatogram of *MSI2* exon 8–*C17orf64* exon 6 fusion.

**Table 1 genes-14-00918-t001:** Mutations identified by DNA sequencing.

Gene	Chr	HGVSC	HGVSP	Varsome Classification	AF
*TP53*	17	c.742C>T	p.(Arg248Trp)	pathogenic	46.85
*ATM*	11	c.6067G>A	p.(Gly2023Arg)	pathogenic	46.31
*ERBB3*	12	c.172G>A	p.(Val58Met)	VUS	7.12
*CSNK1A1*	5	c.702_703insAACATGGAATCA	p.(Ser234_Leu235insAsnMetGluSer)	VUS	15.99
*RPS6KB2*	11	c.358C>T	p.(Arg120Trp)	VUS	54.59

Chr = chromosome. HGVSC = human genome variation society-coding. HGVSP = human genome variation society-protein. VUS = variant of uncertain significance; educational use only. AF = allele frequency.

## Data Availability

The data used and/or analyzed during the current study are available from the corresponding author on reasonable request.

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
