# Peer review of "Detection of a Novel MSI2-C17orf64 Transcript in a Patient with Aggressive Adenocarcinoma of the Gastroesophageal Junction: A Case Report"

_genes, 2023, doi:10.3390/genes14040918_

Round 1

Reviewer 1 Report

This study was planned to understand the genomic landscape of Adenocarcinoma of the esophagus (EAC) and gastroesophageal junction (GEJ-AC), and potentially identify a therapeutic target in a neoadjuvant chemotherapy non-responder 48-year-old man, we adopted a multi-omic approach enabling the simultaneous evaluation of gene rearrangement, mutations, copy number status and microsatellite instability and tumor mutation burden. The authors have faithfully conducted genetic analysis and have thought a lot about it, so I think it is a good paper.

1. FFPE specimen is divided into positive-cohesive one was prevalent (90%) and with local SRC area (<25%), did you select and analyze some of them or did you analyze all of them?, It need to describe in method part.

2. Describe the analysis of only 1 case as a limitation in the Discussion part.

3. If there is a relationship between the gene increase and decrease of MSI2-C17irf64 and the survival rate in KM plotters , etc., it would be a better achievement.

Reviewer 2 Report

This article describes the poor prognosis and treatment resistance of esophageal and gastroesophageal junction cancer. A multi-omic strategy was used to determine potential therapeutic targets in a non-responder to neoadjuvant chemotherapy. The patient showed pathogenic mutations in the TP53 and ATM genes, as well as amplification of the FGFR2 and KRAS genes. Analysis of transcripts uncovered the MSI2-C17orf64 fusion, a previously unreported potential therapeutic target. This study emphasizes the need of conducting in-depth molecular analysis to uncover unique patient-specific indicators for monitoring or addressing disease progression.

Comments-

Here are a few recommendations for enhancing the quality of this research paper:

1-Add further background details: The report should elaborate on the disease (adenocarcinoma of the gastroesophageal junction), the standard treatment options, and the significance of genetic testing in treatment of cancer.

2-Explain the approach: The research could include further information regarding the methodology used for genetic testing, including the quality measures used and the test's limitations. Also, it would be useful to describe how the fusion analysis was conducted and why these specific fusion tools were selected.

3-Add additional interpretation: The report could include greater interpretation of the genetic test findings, particularly addressing the individual mutations identified and their possible effect on the course of cancer and treatment alternatives. 

4-The manuscript should be corrected for errors in grammar, spelling, and punctuation. The wording should be clear and succinct, including explanations of technical words for a wide readership.

Round 2

Reviewer 2 Report

I have no further comments.